# Cessation of Nucleos(t)ide Analogue Therapy in Non-Cirrhotic Hepatitis B Patients with Prior Severe Acute Exacerbation

**DOI:** 10.3390/jcm10214883

**Published:** 2021-10-23

**Authors:** Chia-Yeh Lai, Sheng-Shun Yang, Shou-Wu Lee, Hsin-Ju Tsai, Teng-Yu Lee

**Affiliations:** 1Division of Gastroenterology and Hepatology, Department of Internal Medicine, Taichung Veterans General Hospital, Taichung 40705, Taiwan; student00069@gmail.com (C.-Y.L.); yansh@vghtc.gov.tw (S.-S.Y.); ericest@vghtc.gov.tw (S.-W.L.); a9194024@hotmail.com (H.-J.T.); 2School of Medicine, Chung Shan Medical University, Taichung 40201, Taiwan; 3Ph.D. Program in Translational Medicine, National Chung Hsing University, Taichung 40227, Taiwan; 4Institute of Biomedical Sciences, National Chung Hsing University, Taichung 40227, Taiwan

**Keywords:** hepatitis flare, liver failure, hepatitis B virus, antivirals, discontinuation

## Abstract

Chronic hepatitis B (CHB) with severe acute exacerbation (SAE) is an urgent problem requiring nucleos(t)ide analogue (NA) therapy. We aim to evaluate the clinical relapse (CR) risk after discontinuing NA in patients with prior SAE. Methods: In this retrospective cohort study, CHB patients who discontinued NA therapy were screened between October, 2003 and January, 2019. A total of 78 non-cirrhotic patients who had received NA therapy for CHB with SAE, i.e., bilirubin ≥ 2 mg/dL and/or prothrombin time prolongation ≥3 s, (SAE group) were matched 1:2 with 156 controls without SAE (non-SAE group) by means of propensity scores (age, gender, NA categories, NA therapy duration, and HBeAg status). Results: The 5-year cumulative incidences of severe CR, i.e., ALT > 10X ULN, (42.78%, 95% CI: 27.84–57.73% vs. 25.42%, 95% CI: 16.26–34.58%; *p* = 0.045) and SAE recurrence (25.91%, 95% CI: 10.91–40.91% vs. 1.04%, 95% CI: 0–3.07%; *p* < 0.001) were significantly higher in the SAE group. Prior SAE history (HR 1.79, 95% CI: 1.04–3.06) was an independent factor for severe CR. The 5-year cumulative incidence of HBsAg seroclearance was significantly higher in the SAE group than that in the non-SAE group (16.82%, 95% CI: 2.34–31.30% vs. 6.02%, 95% CI: 0–13.23%; *p* = 0.049). Conclusions: Even though it creates a greater chance of HBsAg seroclearance, NA therapy cessation may result in a high risk of severe CR in non-cirrhotic CHB patients with prior SAE.

## 1. Introduction

There are more than 250 million people worldwide chronically infected with hepatitis B virus (HBV); therefore, the treatment of chronic hepatitis B (CHB) is a huge burden on public health [1]. Although nucleos(t)ide analogue (NA) therapy can reduce the risk of HBV-related cirrhosis and/or hepatocellular carcinoma [2], the ideal endpoint of NA therapy, i.e., hepatitis B surface antigen (HBsAg) seroclearance or seroconversion, is rarely achieved [3]. However, without an effective therapy to eradicate HBV, NA therapy currently remains the standard treatment for patients with CHB. With a concern regarding virological relapse (VR; defined as serum HBV viral load > 2000 IU/mL) or clinical relapse (CR; defined as VR with alanine aminotransferase (ALT) > 2X upper limit of normal (ULN)) after NA therapy cessation, indefinite NA therapy for cirrhotic patients has been recommended in the clinical guidelines [4,5,6]. However, in non-cirrhotic patients, long-term NA therapy may raise several concerns in the real world, including its safety, cost, drug resistance, and patient compliance. Therefore, the alternative endpoints for discontinuing NA therapy in non-cirrhotic patients have been suggested in the current practice guidelines: persistent undetectable serum HBV DNA in hepatitis B e antigen (HBeAg)-negative patients or HBeAg seroconversion in HBeAg-positive patients. Unfortunately, most patients will experience a VR, so NA therapy retreatment for CR is common after cessation of NA therapy [7,8].

Previous systemic review studies have shown that as high as 70% and 50% of patients would experience VR and CR within one year after NA therapy cessation, respectively [9]. Moreover, some patients may suffer from severe CR, hepatic failure, and death [10,11]. However, even though there may be potential harm resulting from VR or CR, benefits from a higher possibility of HBsAg seroclearance after NA therapy cessation has also been reported (6-year cumulative incidence: 13%) [12,13]. The harms and benefits resulting from NA therapy cessation should be well balanced.

Severe acute exacerbation (SAE) of CHB, defined as severe hepatitis B flare with jaundice and/or coagulopathy, is an urgent clinical condition with a significant mortality risk [14] which requires immediate NA treatment [15]. However, the data regarding severe CR, and even SAE recurrence after NA therapy cessation in patients who ever experienced CHB with SAE, remains very limited. In previous studies involving small case numbers, a high rate of SAE recurrence (18–50%) among CHB relapsers has been observed [16]. In a larger case series reported from Taiwan, the cumulative incidence of SAE recurrence in non-cirrhotic patients was 17.2% within a median follow-up duration of 90 months, after discontinuing a 12-month course of lamivudine (LAM) therapy amongst patients who recovered from prior SAE [17]. However, without a well-matched control arm involving patients without SAE, the rates of various clinical outcomes cannot be well compared between patient groups with or without SAE. We therefore aimed to conduct a cohort study for the purpose of evaluating the risk of severe CR after NA therapy cessation in non-cirrhotic CHB patients with prior SAE, through the use of a matched control group.

## 2. Materials and Methods

### 2.1. Study Design

This retrospective cohort study was conducted at Taichung Veterans General Hospital, a tertiary referral center in central Taiwan. All patients who had received NA therapy for CHB were screened between 1 October 2003 and 31 January 2019. Use of NA therapy involving entecavir (ETV), tenofovir disoproxil fumarate (TDF), LAM, and telbivudine (LdT) was permitted during the study period. This study has been approved by the Institutional Review Board of Taichung Veterans General Hospital (CE18314A).

### 2.2. Study Cohort

The patient selection process is shown in Figure 1. We screened 1838 non-cirrhotic patients who had discontinued NA therapy due to achieving the therapeutic endpoints of CHB (undetectable HBV DNA and negative HBeAg). The therapeutic endpoints of CHB were based on the clinical practice regulations of the Taiwan’s National Health Insurance, which were modified according to the Asian-Pacific clinical practice guidelines (HBeAg loss with NA consolidation therapy for 6–12 months amongst HBeAg-positive patients; undetectable HBV DNA with NA consolidation therapy for 12 months amongst HBeAg-negative patients) [5]. The exclusion criteria were as follows: (1) HBsAg seroclearance during NA therapy; (2) loss to follow-up in our hospital after discontinuing NA therapy; (3) a diagnosis of hepatitis C virus (HCV) or human immunodeficiency virus (HIV) co-infection; (4) alcohol abuse; (5) hepatitis other than CHB; (6) liver cirrhosis, which was diagnosed by image studies and supplementary clinical/laboratory features during NA therapy; (7) active malignancy; (8) organ transplantation; and (9) immunosuppressant use. After excluding the above-mentioned confounding conditions, we further divided patients into the SAE and non-SAE cohorts according to their initial indications for starting NA therapy. SAE of CHB was defined as hepatitis B flare (HBV viral load > 2000 IU/mL and ALT > 5X ULN) with jaundice (total bilirubin ≥ 2 mg/dL), and/or coagulopathy (prothrombin time (PT) prolongation ≥ 3 s) [18]. The ULN of ALT was defined according to the updated American Association for the Study of Liver Diseases criteria (>25 U/L for females and >35 U/L for males). After excluding patients with confounding conditions patients in the SAE group were randomly matched 1:2 with patients in the non-SAE group by means of propensity scores, which consisted of age, gender, NA categories, NA therapy duration, and HBeAg status prior to NA therapy. LAM and LdT were categorized as the same group due to their similar characteristics [4,5].

### 2.3. Outcome Measurement

Patients in the SAE and non-SAE groups were followed up from the dates of NA therapy cessation (as the index dates). Before NA cessation, the risk (symptoms/signs) of hepatitis B recurrence was informed. After NA therapy cessation, patients were followed up with liver function tests every 4 weeks in the first 12 weeks, and then every 12 weeks apart [5]. Once CR was noticed, patients would be monitored with a closer follow-up duration. In addition, severe hepatitis flare could be a pre-stage before SAE development, and NA therapy might be initiated to avoid SAE. Therefore, we defined “severe CR”, i.e., a serum ALT > 10X ULN with an HBV viral load > 2000 IU/mL [19], as the major measured outcome in this study. Patients with CR might receive NA retreatment according to the clinical practice regulations of the Taiwan’s National Health Insurance’s regulation: For HBeAg-positive non-cirrhotic patients, HBV viral load should be >20,000 IU/mL, and ALT should be >5X ULN or persistently elevated (>2X ULN for more than 3 months). For non-cirrhotic HBeAg-negative patients, HBV viral load should be >2000 IU/mL, with persistently elevated ALT (>2X ULN for more than 3 months). For patients with SAE NA therapy can be prescribed immediately. Patients who died of hepatic failure were recorded. In addition, HBsAg seroclearance was also recorded as the secondary endpoint.

### 2.4. Statistical Analysis

In order to compare the demographic data of the two study groups, continuous variables were compared using the Mann–Whitney U test, with categorical variables being compared using the Chi-Square test or Fisher’s exact test when appropriate. Through use of a logistic regression model, propensity score analysis was performed in order to examine the comparability of the two study groups. Cumulative incidences of study events were calculated using the Kaplan–Meier method, with the differences in the full time-to-event distributions between the two study groups being compared by a log-rank test. Kaplan–Meier and univariate Cox regression were both used to assess the relationship of various variables to the event. Significant variables were included in multivariate Cox regression models. Analyses were performed using the Statistical Package for the Social Science (IBM SPSS version 22.0; International Business Machines Corp., New York, NY, USA).

## 3. Results

### 3.1. Study Subjects

As shown in Figure 1, 78 non-cirrhotic patients who had received NA therapy for CHB with SAE (the SAE group) were finally matched with 156 non-cirrhotic patients without SAE (the non-SAE group) for analysis. Table 1 shows the demographic characteristics of the two study groups. Age, gender, body mass index (BMI), and diabetes patients were similar in the two study groups. Diabetes was identified due to its potential risk in liver disease progression [20]. Before beginning NA therapy, patients in the SAE group had suffered from significantly longer PT prolongation and higher aspartate aminotransferase (AST), ALT, and bilirubin levels as compared to those in the non-SAE group. However, their serum HBV viral load was not significantly different. In addition, both the NA therapy duration and the NA consolidation duration were also similar. At the time of NA cessation the baseline characteristics were basically similar, although the median ALT level was higher within ULN in the non-SAE group. The degree of liver fibrosis was evaluated by using the Fibrosis-4 index (FIB-4) [21], and the values (median 1.1) were similar in the two study groups. The follow-up duration after cessation of NA therapy was not significantly different.

### 3.2. Clinical Relapse and NA Retreatment

As shown in Appendix A, the 5-year cumulative incidences of CR in the SAE group were not significantly higher than those in the non-SAE group. The 1-year, 3-year, and 5-year cumulative incidences of CR in the SAE and non-SAE groups were 23.51% (95% CI: 13.69–33.34%) vs. 24.98% (95% CI: 17.98–31.97%); 53.88% (95% CI: 40.80–66.96%) vs. 48.00% (95% CI: 39.28–56.72%); and 63.89% (95% CI: 50.45–77.33%) vs. 56.30% (95% CI: 46.71–65.89%), respectively. As shown in Appendix A, the 5-year cumulative incidences of NA retreatment in the SAE group were also not significantly higher than those in the non-SAE group. The 1-year, 3-year, and 5-year cumulative incidences of NA retreatment in the SAE and non-SAE groups were 17.94% (95% CI: 9.07–26.80%) vs. 15.05% (95% CI: 9.24–20.85%); 40.77% (95% CI: 28.23–53.31%) vs. 32.20% (95% CI: 24.10–40.31%); 52.25% (95% CI: 37.94–66.57%) vs. 41.33% (95% CI: 31.85–50.80%), respectively.

### 3.3. Severe Clinical Relapse and SAE Recurrence

As shown in Figure 2, the 5-year cumulative incidences of severe CR in the SAE group were significantly higher than those in the non-SAE group. The 1-year, 3-year, and 5-year cumulative incidences of severe CR in the SAE and non-SAE groups were 15.69% (95% CI: 7.13–24.25%) vs. 11.17% (95% CI: 5.99–16.35%); 32.45% (95% CI: 20.22–44.68%) vs. 19.63% (95% CI: 12.52–26.75%); 42.78% (95% CI: 27.84–57.73%) vs. 25.42% (95% CI: 16.26–34.58%), respectively (*p* = 0.045). In addition, as shown in Figure 3, the 5-year cumulative incidences of SAE recurrence in the SAE group were significantly higher than those in the non-SAE group. The 1-year, 3-year, and 5-year cumulative incidences of SAE recurrence in the SAE and non-SAE groups were 8.83% (95% CI: 2.04–15.62%) vs. 0% (95% CI: 0–0%); 17.42% (95% CI: 6.24–28.60%) vs. 1.04% (95% CI: 0–3.07%); 25.91% (95% CI: 10.91–40.91%) vs. 1.04% (95% CI: 0–3.07%), respectively (*p* < 0.001). However, only one patient died of hepatic failure in each study group.

### 3.4. Multivariable Regression Analysis for Severe Clinical Relapse

As shown in Table 2, after adjusting for the covariances (SAE or non-SAE group, age, gender, BMI, overweight, obesity, diabetes, HBV viral load, PT prolongation, HBeAg positivity, NA therapy duration, NA consolidation duration, NA consolidation > 36 months, NA categories, pre-treatment and post-treatment bilirubin, AST and ALT), male gender (hazard ratio (HR) 2.31, 95% CI: 1.13–4.75; *p* = 0.022), and SAE-group (HR 1.79, 95% CI: 1.04–3.06; *p* = 0.034), were all independent factors for severe CR. Overweight and obesity were, respectively, defined as BMI ≥ 23 kg/m^2^ and BMI ≥ 25 kg/m^2^ [22].

### 3.5. HBsAg Seroclearance

As shown in Figure 4, the 5-year cumulative incidences of HBsAg seroclearance in the SAE group were significantly higher than those in the non-SAE group. The 1-year, 3-year, and 5-year cumulative incidences of HBsAg seroclearance in the SAE and non-SAE groups were 1.45% (95% CI: 0–4.27%) vs. 0% (95% CI: 0–0%); 7.94% (95% CI: 0–17.04%) vs. 0.88% (95% CI: 0–2.61%); 16.82% (95% CI: 2.34–31.30%) vs. 6.02% (95% CI: 0–13.23%), respectively (*p* = 0.049).

## 4. Discussion

In this study, we first demonstrated that although the rates of CR could be similar between the SAE and non-SAE groups, the rates of severe CR, including SAE recurrence, were found to be significantly higher in the SAE group. In multivariable regression analysis, a patient’s SAE history was an independent factor for severe CR. Our study’s findings clearly raised the concern of severe CR occurring after NA cessation in patients having a history of CHB with SAE. Interestingly, higher rates of HBsAg seroclearance, one of the benefits of NA therapy cessation, was seen in the SAE group. Although the most concerning clinical issue to handle after NA therapy cessation is severe CR, which may result in liver decompensation and ultimately death, a higher rate of HBsAg seroclearance may raise the argument that NA therapy should be discontinued even under a high severe CR rate in patients with prior SAE history. Therefore, the pros and cons of NA therapy cessation should be well balanced, and an accurate outcome predictor, such as HBV ccc-DNA-correlated biomarker, should be further studied during future research.

A high rate of CR is a problem after NA therapy cessation. CR rates could be as high as 51–82% in HBeAg-positive seroconversion patients within 4 years [23,24] and 41.9–63.2% in HBeAg-negative patients within one year [25,26,27]. Even so, with a low rate of liver decompensation, NA therapy cessation has been advocated under the considerations of a higher HBsAg seroclearance rate, less adverse effects of long-term NA therapy, and lower costs. However, as the data in this study shows, the degree of CR was more severe in the SAE group than that in the non-SAE group, and the significantly higher risk of severe CR and SAE recurrence may raise the concern of potential mortality risk if there is a lack of prompt clinical monitoring after NA therapy cessation. In addition, even though we did not discover significantly different mortality rates while all patients were being closely followed up in this study, patients with SAE recurrence may still suffer from severe liver injury and fibrosis progression. With a high risk of severe CR, the decision regarding cessation of NA therapy should not be easily made, and indefinite NA therapy may therefore be a safer choice.

Compatible with the data of previous studies, the risk of severe CR was quite high after NA therapy cessation in patients who had ever experienced CHB with SAE. Chang et al. reported that 5-year cumulative incidences of SAE recurrence after cessation of 12-month LAM therapy were 18.1% and 21.2% in non-cirrhotic and cirrhotic patients, respectively, of those who had recovered from hepatitis B flare with prior SAE [17]. They stated a hypothesis that if the patients with SAE received a longer and more potent NA therapy, their relapse rate would be lower. However, even though patients in this study received a stronger NA therapy (mostly ETV or TDF) over a median duration of 36 months, the SAE recurrence rate remained as high as 25.9% over the course of 5 years. Therefore, the treatment duration and potency of NA may not be the major factors associated with SAE recurrence.

In the era prior to NA therapy, the rate of spontaneous SAE in CHB patients could be as high as 10–30% every year [28]. The mechanisms of SAE may be explained by multiple factors, including virological genotype [16], virological mutations [29], or immune cell dysfunction [30]. As shown in the multivariable regression analysis in this study, SAE history was an independent factor for severe CR; however, the reasons why patients with a history of SAE were more likely to develop SAE recurrence remain unknown. In this study, most SAE recurrence occurred within 2 years after NA therapy cessation; therefore, the recurrence of a VR-triggered immune response should be considered. However, even with similar CR rates the chance of severe CR was much lower in the non-SAE group and a host factor should have been involved. Further studies to investigate the predisposing factors for SAE recurrence are still required.

HBsAg seroclearance or seroconversion is an ideal endpoint in the treatment of CHB, but is rarely achieved. Previous studies have reported that NA therapy cessation may result in a higher rate of HBsAg seroclearance. The restoration of the HBV-specific T-cell immune function may be one of the explanations for this [31]. However, the rate was much lower in the Asian population compared to that of Caucasians [32]. In our study, the 5-year cumulative incidences of HBsAg seroclearance were significantly higher in the SAE group compared to that of the non-SAE group, and a stronger immune response in patients who experienced SAE may be one of the explanations. A higher rate of HBsAg seroclearance may be an important reason for advocating NA therapy cessation; however, patients in the SAE group would take on another high risk for SAE recurrence.

Several limitations should be acknowledged with regard to this study. First, some unexpected bias may exist in this retrospective study; however, we have excluded or matched possible confounding factors between the two study groups to minimize the risk for any study bias. Second, the sample size in this single center study was not large. Furthermore, some other important clinical outcomes, such as hepatocellular carcinoma development [33], cannot be well evaluated. However, the incidence of CHB with SAE is not high in the era of NA therapy, and this study has been one which involved bigger sample sizes. A multi-center study could be encouraged in order to confirm our findings. Third, the data of some novel CR predictors, such as quantitative HBsAg (qHBsAg), could not be completely obtained in this retrospective study. Although previous studies reported that a higher level of qHBsAg may be related to a higher CR rate [28], the rates of CR were similar between the SAE and non-SAE groups in the present study (Appendix A). However, even with similar VR and CR rates, the chance of severe hepatitis relapse remained much higher in the SAE group. The blood level of qHBsAg might not play a central role in severe CR. A prospective study should be conducted in order to reveal any new risk factors. Fourth, although the duration of NA consolidation therapy was not an independent factor of severe CR in our analysis, the median NA consolidation duration was not very long (26 months). Therefore, the risk of severe CR after long-term NA therapy should be further studied. However, our study clearly demonstrated that severe CR after cessation of NA therapy should be carefully noticed in patients with prior SAE history. Fifth, although the FIB-4 index values were small (median 1.1) in the two study groups at the time of NA cessation, the liver stiffness should have been changed after a period of NA therapy [34]. However, when patients were suffering from CHB with SAE, the data regarding liver stiffness could be inaccurate due to severe hepatitis. Therefore, for avoiding NA cessation in cirrhotic patients, the degree of liver stiffness should be carefully evaluated after the episode of SAE.

## 5. Conclusions

In conclusion, even though having a higher chance of HBsAg seroclearance, the risk of severe CR, including SAE recurrence, is very high after NA therapy cessation in non-cirrhotic patients who have experienced CHB with SAE. Therefore, prior to making any decision to discontinue NA therapy, the high risk of severe CR should be well-considered and closely monitored.

## Figures and Tables

**Figure 1 jcm-10-04883-f001:**
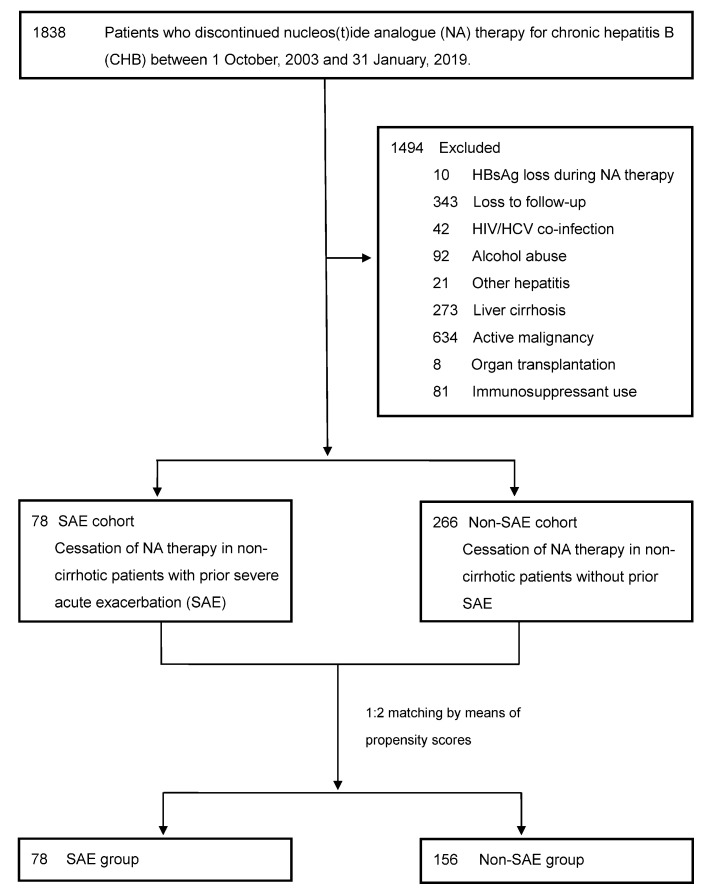
Selection of study subjects. HBsAg, hepatitis B surface antigen; HCV, hepatitis C virus; HIV, human immunodeficiency virus; SAE, severe acute exacerbation.

**Figure 2 jcm-10-04883-f002:**
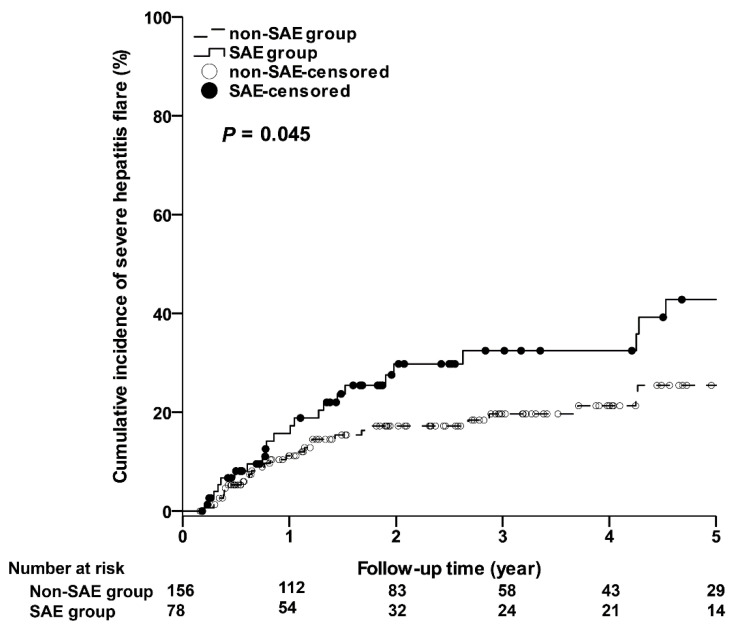
The cumulative incidences of severe clinical relapse. SAE, severe acute exacerbation.

**Figure 3 jcm-10-04883-f003:**
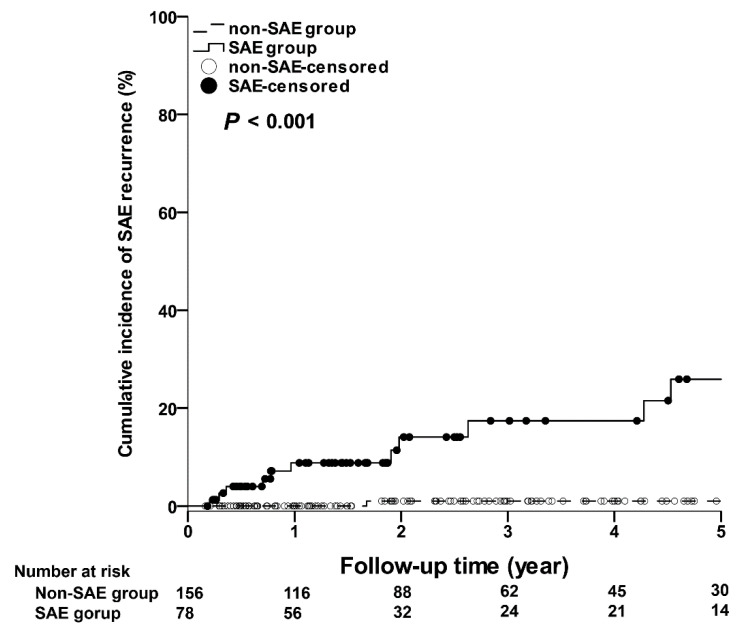
The cumulative incidences of severe acute exacerbation recurrence. SAE, severe acute exacerbation.

**Figure 4 jcm-10-04883-f004:**
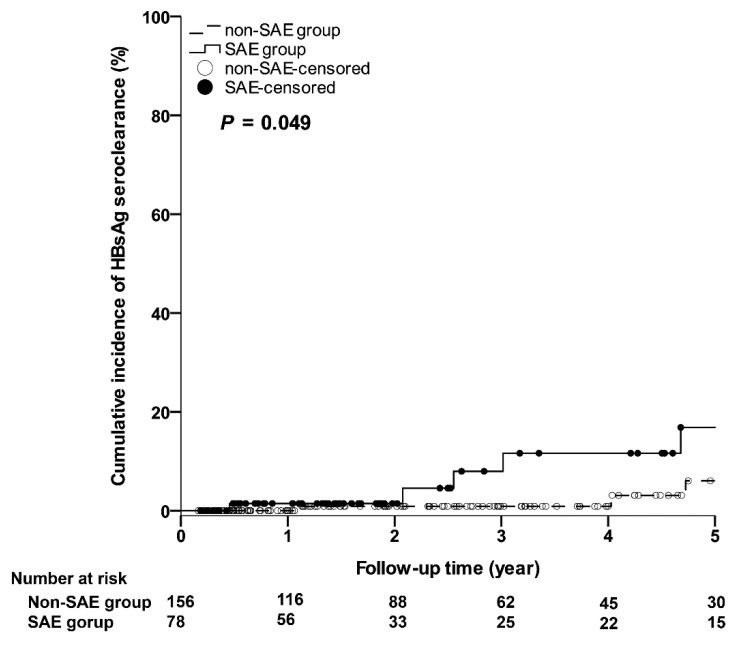
The cumulative incidences of HBsAg seroclearance. HBsAg, hepatitis B surface antigen; SAE, severe acute exacerbation.

**Table 1 jcm-10-04883-t001:** Baseline characteristics of study subjects.

Characteristics	Non-SAE Group	SAE Group	
	*n* = 156	*n* = 78	*p* Value
Age, years	43 (35–53)	42 (31–53)	0.650
Male, *n* (%)	112 (71.8%)	56 (71.8%)	1.000
BMI, kg/m^2^	24.5 (22.8–27.1)	23.7 (22.1–25.8)	0.114
Diabetes, *n* (%)	20 (12.8%)	12 (15.4%)	0.737
Data when NA therapy initiation			
HBV DNA, log^10^ IU/mL	6.21 (5.11–7.36)	6.68 (5.02–7.91)	0.359
Bilirubin, mg/dL	0.8 (0.6–1.1)	5.2 (2.7–10.1)	<0.001 **
PT prolongation, second	0 (0–0.3)	2.4 (0.25–5.35)	<0.001 **
AST, ULN	3.7 (2.5–7.3)	32.5 (14.8–49.5)	<0.001 **
ALT, ULN	8.5 (4.8–15.9)	41.5 (22.9–67.9)	<0.001 **
HBeAg positivity, *n* (%)	62 (39.7%)	25 (32.1%)	0.315
NA therapy duration, month	36.0 (35.0–38.0)	36.0 (22.9–36.0)	0.454
NA consolidation duration, month	26.04 (14.46–30.51)	24.95 (15.59–31.53)	0.779
NA therapy categories			0.468
Entecavir, *n* (%)	67 (42.9%)	31 (39.7%)	
Tenofovir, *n* (%)	27 (17.3%)	10 (12.8%)	
Lamivudine or telbivudine, *n* (%)	62 (39.7%)	37 (47.4%)	
Data when NA therapy cessation			
Bilirubin, mg/dL	0.7 (0.5–0.8)	0.7 (0.5–0.9)	0.117
AST, ULN	0.7 (0.6–0.9)	0.7 (0.6–0.8)	0.816
ALT, ULN	0.8 (0.6–1.1)	0.7 (0.5–0.9)	0.021 *
PLT, 10^9^/L	202.0 (177–249)	206.0 (168–251)	0.879
FIB-4	1.1 (0.7–1.6)	1.1 (0.7–1.4)	0.843
Follow-up duration, month	24.87 (10.28–47.54)	19.56 (9.2–37.17)	0.229

* *p* < 0.05, ** *p* < 0.01. ALT, alanine aminotransferase; AST, aspartate aminotransferase; BMI, body mass index; FIB-4, fibrosis-4 index; HBeAg, hepatitis B e antigen; HBV, hepatitis B virus; NA, nucleos(t)ide analogue; PLT, platelet; PT, prothrombin time; SAE, severe acute exacerbation; ULN, upper limit of normal.

**Table 2 jcm-10-04883-t002:** Cox proportional hazards regression model analysis for severe clinical relapse.

Variables	Univariable		Multivariable	
	HR (95% CI)	*p* Value	HR (95% CI)	*p* Value
SAE vs. Non-SAE	1.76 (1.03–3.01)	0.040 *	1.79 (1.04–3.06)	0.034 *
Age, year	1.01 (1.00–1.03)	0.143		
Male vs. female	2.27 (1.11–4.66)	0.025 *	2.31 (1.13–4.75)	0.022 *
BMI, kg/m^2^	0.96 (0.87-1.05)	0.369		
Overweight vs. non-overweight ^a^	0.96 (0.87-1.05)	0.369		
Obesity vs. non-obesity ^b^	0.71 (0.37-1.37)	0.309		
Diabetes vs. Non-diabetes	1.66 (0.83–3.32)	0.149		
Data when NA therapy initiation				
HBV DNA, log^10^ IU/mL	1.15 (0.93–1.42)	0.186		
Bilirubin, mg/dL	1.03 (0.99–1.08)	0.146		
PT prolongation, second	1.05 (0.95–1.16)	0.313		
AST, ULN	1.00 (0.99–1.01)	0.913		
ALT, ULN	1.00 (0.99–1.01)	0.602		
HBeAg positivity	0.66 (0.37–1.17)	0.154		
NA therapy duration, month NA consolidation duration, m NA consolidation > 36 months	1.01 (1.00–1.02) 1.01 (0.99–1.03) 1.17 (0.36–3.82)	0.143 0.524 0.794		
ETV vs. other NAs	0.95 (0.55–1.66)	0.869		
Data when NA therapy cessation				
Bilirubin, mg/dL	2.31 (0.41–12.85)	0.340		
AST, ULN	0.69 (0.27–1.79)	0.450		
ALT, ULN	0.98 (0.71–1.36)	0.904		
PLT, 10^9^/L	0.99 (0.99–1.00)	0.095		
FIB-4	1.09 (0.73–1.63)	0.682		

* *p* < 0.05. ^a^ Overweight: BMI ≥ 23 kg/m^2^; ^b^ Obesity: BMI ≥ 25 kg/m^2^. ALT, alanine aminotransferase; AST, aspartate aminotransferase; BMI, body mass index; CI, confidence interval; ETV, entecavir; FIB-4, fibrosis-4 index; HBeAg, hepatitis B e antigen; HBV, hepatitis B virus; HR, hazard ratio; NA, nucleos(t)ide analogue; PLT, platelet; PT, prothrombin time; SAE, severe acute exacerbation; ULN, upper limit of normal.

## Data Availability

Data available on request from the authors.

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
