# Peer review of "Cessation of Nucleos(t)ide Analogue Therapy in Non-Cirrhotic Hepatitis B Patients with Prior Severe Acute Exacerbation"

_jcm, 2021, doi:10.3390/jcm10214883_

Round 1

Reviewer 1 Report

Authors addressed all my comments in the revised manuscript.

Author Response

Thank you very much for your careful review.

Reviewer 2 Report

The topic is undoubtedly of interest as clear "stopping rules" for NA therapy in HBV patients are still matter of debate.

I have the following suggestions:

1) In cumulative curves were there censored patients? if so, this should be reported in the figure.

2) Table 2: i am surprised that when lower confidence interval is 1.00, the result is not significant....could the authors re-check their analysis?

3) It would be nice to assess the eventual role of some clinical features in the regression analysis. For example the role of diabetes (in this regard cite this comprehensive review: PMID: 23845075)

4) Two limitations should be adequately commented in the discussion:

  • Lack of data on the dynamic changes of liver stiffness in these patients (cite and comment briefly the recent meta-analysis on this topic: PMID: 29807871 )
  • The relatively short follow-up that prevents the assessment of long-term consequences for example the incidence of HCC (in this regard cite and briefly comment the paper: PMID: 25085684)

Author Response

Dear Editors and Reviewers,

Thank you very much for allowing us the opportunity to improve our manuscript. We have revised the manuscript according to your constructive comments and valuable suggestions. We are resubmitting it for consideration for publication as an original paper in Journal of Clinical medicine. The following are our point-by-point responses to your comments (changes are presented as Pages and Lines of the copy using the tract change function). 

Reviewer #2:

The topic is undoubtedly of interest as clear "stopping rules" for NA therapy in HBV patients are still matter of debate. I have the following suggestions:

Comment 1: In cumulative curves were there censored patients? if so, this should be reported in the figure.

Response: Thank you for this helpful suggestion. We have revised all the figures with cumulative curves, in which the censored patients are reported (Figure 2, Figure 3, Figure 4 in the manuscript, and Figure S1 and Figure S2 in the supplementary material).

Comment 2: Table 2: I am surprised that when lower confidence interval is 1.00, the result is not significant....could the authors re-check their analysis?

Response: Thank you for your careful review. In Table 2, there are two items with a lower confidence interval demonstrated as 1.00: Age per year (HR 1.01 [1.00-1.03]; p = 0.143) and NA therapy duration per month (HR 1.01 [1.00-1.02]; p = 0.143). We have carefully rechecked these data: Each data is actually smaller than 1.00 (i.e. 0.999), which is the result of rounding off to the 2nd decimal place. We confirm that all the data are current, and we believe that a statistically insignificant p value can explain this condition (Table 2).

Comment 3: It would be nice to assess the eventual role of some clinical features in the regression analysis. For example the role of diabetes (in this regard cite this comprehensive review: PMID: 23845075)

Response: Thank you for this constructive suggestion. We agree with you that diabetes could be a potential risk factor in liver disease progression (Curr Diabetes Rev. 2013 Sep;9(5):382-6); however, the role of diabetes in hepatitis B relapse after NA therapy cessation remains unclear. Therefore, we identified diabetes patients according to the medical records, and the diabetes data have been analyzed in the baseline characteristics of study subjects and the Cox proportional hazards regression model for severe clinical relapse. As shown in Table 1, the patient proportions of diabetes were not significantly different between the non-SAE group and the SAE group (12.8% vs. 15.4%, p = 0.737). In addition, diabetes was not significantly associated with severe clinical relapse in the regression analysis (HR 1.66, 95% CI: 0.83-3.32; p = 0.149). We have revised the related statements and data in the manuscript (Page 3, Lines 140-141; Table 1; Page 5, Line 188; Table 2; Page 11, Reference 20).

Comment 4: Lack of data on the dynamic changes of liver stiffness in these patients (cite and comment briefly the recent meta-analysis on this topic: PMID: 29807871)

Response: Thank you for this useful comment. Although the FIB-4 index values were small (median 1.1) in the two study groups at the time of NA cessation, the liver stiffness should have been changed after a period of NA therapy (Dig Liver Dis. 2018 Aug;50(8):787-794). However, when patients were suffering from CHB with SAE, the data regarding liver stiffness could be inaccurate due to severe hepatitis. Therefore, for avoiding NA cessation in cirrhotic patients, the degree of liver stiffness should be carefully evaluated after the episode of SAE. We have added the related limitations and discussions in the manuscript (Page 10, Lines 294-300; Page 12, Reference 33).

Comment 5: The relatively short follow-up that prevents the assessment of long-term consequences for example the incidence of HCC (in this regard cite and briefly comment the paper: PMID: 25085684)

Response: Thank you for this important comment. We totally agree with you that hepatitis B relapse is not the only concerned clinical outcome in the issue regarding NA therapy cessation, and HCC development or recurrence should be another important issue (Dig Liver Dis. 2014 Nov;46(11):1014-9). However, the sample size in this single center study was not large; therefore, some other important clinical outcomes, such as hepatocellular carcinoma development, cannot be well evaluated. A multi-center study could be encouraged in the future. We have added the related limitations and discussions in the manuscript (Page 9, Lines 278-280; Page 12, Reference 32).

Round 2

Reviewer 2 Report

The manuscript is OK in the current version. Thank you!

Author Response

Thank you very much for your careful review.

This manuscript is a resubmission of an earlier submission. The following is a list of the peer review reports and author responses from that submission.

Round 1

Reviewer 1 Report

In this study, authors evaluated the clinical relapse risk and the rate of HBsAg seroconversion after discontinuing NA therapy in chronic hepatitis B patients with and without prior severe acute exacerbation. This is a well-designed and well-presented study which will be of interest to scientists/clinicians in HBV field. 

A few minor suggestions:

  1. Define clinical relapse and virological relapse in the introduction section to improve readability.
  2. Reformat Table 1 as values do not align with patients characteristics

Reviewer 2 Report

The authors address a clinically very important topic  that encompasses enormous groups of patients worldwide who are on antiviral therapy with nucleoside analogues. They provide provocative information about the risks and outcomes of clinical relapse after d/c of therapy when certain criteria are met . They contribute to a worldwide debate about the pros and cons of

There are various aspects that I think could help to upgrade the manuscript

Major points

  1. The authors state that they aimed to do conduct a cohort study, but then subsequently indicate that they did a retrospective study with groups who had already had SAE and a group (propensity match) that had not. It remains unclear (many patients lost for follow up that therefore were not included) what the exact protocols were that were executed. Healthy folks above all were allowed to be included?
  2. The relevance of point 1 includes that the reader may better understand  why they are informed about numerous laboratory data (that can be abnormal without any symptom) and can easily be retrieved  but that there is a remarkable lack of information about associated  clinical symptoms or lack thereof that were identified (prospectively questioned?) and could potentially help to identify at an early stage those who are developing issues (like we warn patients on INH that if they start to feel tired, note discoloration of their urine etc. they should promptly be evaluated). This is probably in part due to the retrospective nature. It does not necessarily invalidate their observations, but data are therefore in part more superficial.

They may provide the reader with a little more clarification and other than mortality, what was the impact of the CR, for example in clinical admissions because of symptomatic disease etc.

  1. This reader is confused about too many inconsistent or confusing definitions throughout abstract  and manuscript of  severe CR (ALT > 10x , elsewhere 5x, +/- viral load.)  and in the manuscript abstract SAE is not defined. This requires attention. 
  2. Study cohort:  “We screened…..was there a specific protocol in place with interval testing of all patients?
  3. The selection aspects in fig 1 show an enormous number of exclusions: any comments?

  1. Reflecting on the history of outcomes of therapy, not necessarily that you need to address but some thought re discussion:  Closest to the NuclAnalogs  CR were what happened with interferon: Patients with certain characteristics (genotype, base line inflammation etc.) did best and we were always happy to see some inflammatory response reflected in elevation of transaminases: Their seroclearance (HbsAg conversion) rate was highest. In a similar way: significant inflammation (jaundice) tends to be associated with the highest clearance rate. We want something to happen, we want to no overshoot of the system and we do not want a flare as such leading to fibrosis. Is there any evidence that SAE lead to more pronounced scar formation?  The reader would be helped with more specific numbers re known mortality. From the interferon we know it were the cirrhotics who would get fatal flares and this was  likely because  no adequate hepatic reserve. Could there have been an underestimation of underlying liver disease? Do we have any pathological information?

How will the picture change once cccDNA can more effectively be eliminated?

Minor:

Line 12/13 urgent problem that may require…… We did a retrospective evaluation of…..

A total of …..

“This finding may again raise the argument  that NA therapy should be discontinued even under a high CR rate.”  Rephrase, help the reader comprehend what you mean